# Does Property Return Affect Seller Behavior? An Empirical Study of China’s Real Estate Market

**DOI:** 10.3390/bs13010055

**Published:** 2023-01-06

**Authors:** Hongfei Li, Limin Liang, Chengjiu Sun

**Affiliations:** 1School of Economic, Xiamen University, Xiamen 361005, China; 2School of Journalism and Communication, Xiamen University, Xiamen 361005, China; 3School of Economics, Nankai University, Tianjing 300071, China

**Keywords:** real estate market, disposition effect, loss aversion

## Abstract

This paper examines the relationship between property return and seller behavior and aims to test the disposition effect in China’s real estate market. Using transaction data in Beijing, we find that loss properties have a lower sell propensity relative to gain properties, confirming the existence of the disposition effect. We also find that the disposition effect is more pronounced in samples with shorter holding periods. Sellers with financial constraints and popular projects are more likely to show the disposition effect. Furthermore, we find that sellers exhibit loss aversion; specifically, sellers with loss properties are likely to set a higher listing price, which provides indirect evidence for the disposition effect.

## 1. Introduction

The disposition effect, first raised by Shefrin and Statman [1], refers to the phenomenon that investors are less inclined to sell properties with losses relative to the reference price than properties with gains. This disposition effect is a well-known psychological phenomenon and well documented by a series of papers, including Shefrin and Statman [1], Odean [2], Weber and Camerer [3], Genesove and Mayer [4], Locke and Mann [5], Kumar [6], Hong et al. [7], and so on. However, empirical studies on the disposition effect mainly focuses on the stock markets, while few studies have been conducted on the real estate market. It is doubtful whether stock trading can have an impact on consumption due to equity assets being a relatively small percentage of household assets, while real estate is large enough to exert an influence on consumption (Hong et al. [7]). Case et al. [8] found that house prices could have a more notable impact on household consumption than equity assets. This situation is even more pronounced in China, where real estate accounts for 70% to 85% of household wealth (Huang [9]; Xie and Jin [10]); it is the most important and largest component of Chinese household wealth and has a greater and wider impact on the economy.

Compared with stock markets, a challenge of studying the disposition effect in the real estate market lies in calculating the unrealized return (*gain* or *loss*) during the holding period. On the one hand, there is a lack of information on the purchase price, which is a reference point commonly used to define the gain or loss of an asset (Odean [2]; Genesove and Mayer [4]; Ben-David and Hirshleifer [11]; Hong et al. [7]). However, the transaction records concerning second-hand housing that are currently available do not involve the initial purchase price from the seller in China. On the other hand, stock prices are available for any period of time, while housing prices are only recorded after the houses are sold. This paper applies a unique dataset and methodology to overcome these difficulties. We combine two datasets: the listing data and the historical transaction data. The listing dataset records the purchase time of the property listed for sale, as well as the listing price and the listing time, while the historical transaction dataset includes transaction information, such as the transaction price and transaction time. By matching these two datasets, we are able to estimate the purchase price and the historical potential price during the holding period, as well as the unrealized return, for each listed property.

We find that compared with *gain* property, a *loss* property has a lower propensity to sell, which confirms the existence of the disposition effect in China’s real estate market. We also find that the disposition effects are more pronounced in samples with a holding period shorter than two years, because short-term holdings are more likely to be motivated by investment rather than consumption, which is consistent with the views of Ben-David and Hirsheifer [11] and Hong et al. [7]. After a series of robustness tests, such as limiting the scope of purchase dates, removing unreliable samples and extreme returns, adjusting expected prices, and applying different empirical methods, our results are still robust. The heterogeneity analysis shows that sellers with financial constraints tend to exhibit a more prominent disposition effect, and the disposition effect is more significant in popular projects with high liquidity. Furthermore, sellers with *loss* properties are likely to set a higher listing price and get a higher transaction price. In other words, sellers exhibit loss aversion in real estate transactions, which is consistent with Genesove and Mayer [4] and Engelhardt [12].

This paper has three main contributions: Firstly, previous empirical studies on the disposition effect have mainly focused on the stock market, and there is less research on the disposition effect in the real estate market; thus, this paper provides evidence of the disposition effect in the Chinese real estate market. Secondly, this paper provides a new empirical method to test the effect of the holding period on the propensity to sell, which can provide new ideas for research related to the propensity to sell houses. Thirdly, the findings of this paper help to understand the behavior and psychology of property investors and provide direction for policy makers to limit property speculation and stabilize the housing market.

This paper is organized as follows: Section 2 introduces the data and main variables and describes the empirical models. Section 3 presents the baseline empirical results. Section 4 presents further analysis. Finally, discussion are presented in Section 5 and conclusions in Section 6.

## 2. Data and Research Methodology

### 2.1. Data

We collect data from one of the largest second-hand housing trading platforms in Beijing. There are mainly two types of data: the historical transaction data and the listing data.

The historical transaction data cover a timespan from the first quarter of 2012 to the fourth quarter of 2020. By eliminating the transaction records with key information missing (such as transaction price and housing size) and winsorizing the transaction price (Winsorizing in this paper refers to deleting samples with a transaction price of less than 10,000 yuan per meter square and samples with a transaction price that exceeds 200,000 yuan per meter square), we obtain more than 820,000 transaction records concerning 3623 residential projects. Each record comprises transaction information (including the listing price, transaction price, listing date, and transaction date) and house characteristics (the number of bedrooms, living rooms, and bathrooms and the orientation and floor of the house). The definitions and descriptive statistics of the main variables contained in the historical transaction data are given in Table 1. Table 1 shows that the average listing price is 49,780 yuan/m^2^, and the transaction price is 48,530 yuan/m^2^. Clearly there is a price reduction of 1350 yuan/m^2^ from the listing price to the transaction price. Generally, the transaction price closely matches the market value of a house, while the listing price can reflect the seller’s psychological expectations. For example, Genesove and Mayer [4] argued that sellers subject to expected losses set higher listing prices because of loss aversion. Engelhardt [12] also showed that *loss* sellers were more reluctant to sell at lower prices. Therefore, this paper will adopt the transaction price when estimating the value and return of a house. It can also be seen from Table 1 that the average area of the houses is 84.66 square meters, and the most common layout is 2 bedrooms, 1 living room, and 1 bathroom. Furthermore, during the sample period, the average time-on-the market (the interval between transaction date and listing date) is 76 days, which can reflect the liquidity of the market (Giglio et al. [13]). 

Regarding the listing data, all listing records are from 1 January 2019 to 31 December 2020. Compared with the historical transaction data, the transaction price is missing in the listing dataset. We are able to get the transaction price of some listing records by matching the house identification codes from the two datasets. At the same time, the listing data contain two more variables: purchase date and mortgage information (if the house is mortgaged). The purchase date is a key variable because the purchase price is typically used as a reference point for calculating unrealized return. This paper determines the purchase price based on the historical transaction price and the purchase date.

We do not use all of the data from the listing dataset. First, we match 3623 projects by the project names in the historical transaction dataset and only retain the data of the matching projects. Then, we delete the listing records with key information missing (such as purchase date). Finally, we delete the listing records with a date of purchase before the year 2017, mainly out of two considerations: firstly, for earlier dates of purchase and longer holding time periods, investors are more likely to trade for consumption rather than speculative motive (Ben-David and Hirsheifer [11]; Hong et al. [7]); secondly, led by a series of stimulus policies from 2015 to 2016, Beijing’s housing prices have surged, and almost all sellers will yield gains when selling a house purchased before 2017 in 2019. The dramatic upward trend in the real estate market will lead to invalid testing of the disposition effect (Odean [2]). Given the stable horizontal price movement and divergence of housing prices in Beijing after 2017, this period is more conducive to testing the disposition effect. We finally acquire nearly 30,000 records. Table 2 presents the main variables of listing records and their descriptive statistics. To be noted, we calculate the distance from each residential area to Tiananmen Square (city center) according to their latitude and longitude to control the location information.

### 2.2. Research Methodology

The main purpose of this paper is to examine whether a *loss* property has a low to sell propensity. As return (*gain* or *loss*) and sell propensity are the two key variables, next we first introduce the design of these two variables, and then present the empirical design.

#### 2.2.1. Return

To calculate the property return, the first thing is to determine the purchase price. Our data do not contain the purchase price of the listed property, but the purchase date is available. In order to estimate the purchase price, we first estimate the historical potential price of the house. We do not use the historical average price of the project as the historical potential price but refer to Hong et al. [7] to estimate the historical potential prices of different housing types in the same project, as shown in Equation (1). In this way, the price differences of houses with different attributes can be preserved. Since we estimate the hedonic models (Equation (1)) project by project, we only control house characteristics including *Size*, *Level1*, *Level2*, *Floor*, *Bedroom*, *Livingroom*, *Bathroom* and *South*. We do not need to control project level characteristics such as *Distance*. Table 3 presents the descriptive information of the 3623 estimates.
(1)$$\begin{array}{l} \log price_{i,t}=\sum_{t=2012Q1}^{2020Q4}\beta_{t}Quarter_{t}+\theta_{1}Size_{i}+\theta_{2}Level1_{i}+\theta_{3}Level2_{i}+\theta_{4}Floor_{i}\\ \theta_{5}Bedroom_{i}+\theta_{6}Livingroom_{i}+\theta_{7}Bathroom_{i}+\theta_{8}South_{i}+\varepsilon_{i,t} \end{array}$$

Table 3 shows that the coefficients of house size and higher floor (*Level2*) are both negative, indicating that larger and higher-floor houses will be subject to price discount, while middle floors (*Level1*) and south facing houses with more rooms such as bedrooms, living rooms, and bathrooms will have a price premium. Moreover, the mean value of *R^2^* of 3623 regression equations has hit 0.907, and the 10% quantile has reached 0.792, which shows that the hedonic price model has good explanatory power, and the set of variables is a full representation of the market value. In addition, other unobserved variables have a limited impact.

The quarterly dummy variable coefficient, estimated from Equation (1), can be used to capture the historical price at the project level. The historical potential price $\log\hat{p}rice_{i,t}$ of a house in the project can be estimated by:(2)$$\log\hat{p}rice_{i,t}={\hat{\beta }}_{t}+\hat{\theta }X_{i}$$
where ${\hat{\beta }}_{t}$ is the quarterly dummy variable estimated by Equation (1) and $X_{i}$ denotes house attributes. Since some projects have no transactions in certain seasons and ${\hat{\beta }}_{t}$ is missing, we use linear interpolation to fill in the missing values. Figure 1 shows the average historical potential price change of 3623 neighborhoods estimated by Equation (2) and the price change in the second-hand housing market published by the Beijing Bureau of Statistics. The two curves vary in value but share the same changing trend before 2016, but they are the same in both value and changing trend after 2016, which proves the representativeness of the samples in this paper and the reliability of the estimation method.

The calculation of unrealized return (*gain* or *loss*) depends on the reference point (initial purchase price). Compared with the stock market, one of the advantages of studying the disposition effect in the real estate market is that the purchase price is unique, and there is no need to calculate the reference price based on a weighted average price because the stock market needs to adjust the account (additional purchase or partial sale) according to the seller. In this paper, the purchase price is calculated according to the purchase date of the house and Equation (2), and the purchase price is used as a reference point to define the unrealized return of the house during the holding period, as follows:(3)$$R_{i,t|t_{0}}=\log\hat{p}rice_{i,t}-\log\hat{p}rice_{i,t_{0}}={\hat{\beta }}_{t}-{\hat{\beta }}_{t_{0}}$$
where $t_{0}$ represents the purchase date of house *i*, $R_{i,t|t_{0}}$ represents the unrealized return of house *i* during the purchase period $t_{0}$ and holding period t. For example, if a house was purchased in the first quarter of 2018 (2018Q1), and listed for sale in the fourth quarter of 2019 (2019Q4), there would be seven unrealized returns. 

#### 2.2.2. Sell Propensity

Next, we define the sell propensity, $Sell_{i,t}$, as follows:(4)$$Sell_{i,t}=\{\begin{array}{cc} 1, & \begin{array}{cccc} Listed & in & period & t+1 \end{array}\\ 0, & Otherwise \end{array}.$$

$Sell_{i,t}$ indicates that the houseowner decides whether to sell the house in the next period by observing the current housing price, but the houseowner may also make judgments based on the expected price of the next period. In the robustness analysis, we examine the impact of the expected price.

According to Equation (4), we can derive multiple pieces of data (as shown in Table 4) from a transaction record of the house *i* sold after the holding period t. Specifically, the propensity to sell the house held by the investor from period 0 to period t−1 is 0, and the corresponding potential returns during the period are $R_{0|t_{0}},R_{1|t_{0}},\cdots ,R_{t-1|t_{0}}$. When the investor decides to sell the house in the next period at t, the value is 1 and the housing attributes $X_{i}$ remains unchanged in all periods.

#### 2.2.3. Empirical Design

Finally, we introduce the empirical model of testing the disposition effect:(5)$$\mathrm{Pr}(Sell_{i,h,t}=1)=\Phi (\alpha_{0}+\alpha_{1}Loss_{i,h,t}+\alpha_{2}X_{i}+\alpha_{3}Z_{i,h,t}+\eta_{h}+\varphi_{t}+\varepsilon_{i,h,t})$$
(6)$$Sell_{i,h,t}=\alpha_{0}+\alpha_{1}Loss_{i,h,t}+\alpha_{2}X_{i}+\alpha_{3}Z_{i,h,t}+\eta_{h}+\varphi_{t}+\varepsilon_{i,h,t}.$$

Equations (5) and (6) are the Probit model and the OLS model, respectively, where: (7)$$Loss_{i,h,t}=\{\begin{array}{cc} 1 & R_{i,h,t}<0\\ 0 & Otherwise \end{array}.$$

$X_{i}$ are the housing attributes that do not change with time, as listed in Panel A of Table 3. $Z_{i,h,t}$ represents other control variables, including if the house is mortgaged, the distance to Tiananmen Square, and the holding period. In order to control the investor’s financial constraints, the total house purchase price processed by logarithm is included in $Z_{i,h,t}$. Moreover, we control the liquidity condition of the project. The construction method of the liquidity condition is to first calculate the average transaction cycle $\bar{TOM}$ of the project in period *t* according to the historical transaction data, and then take a logarithm to get LnTOM. $\eta_{h}$ and $\varphi_{t}$ represent the region and time fixed effects, respectively.

## 3. Empirical Results

### 3.1. Baseline Results

According to two considerations, we need to conduct a sub-sample test in agreement with the holding period. On the one hand, real estate with different holding periods may be motivated by different transaction motives, specifically, real estate with shorter holding periods is more likely to be so out of investment rather than consumption motives (Ben-David and Hirsheifer [11]; Hong et al. [7]). On the other hand, the transaction costs and taxes involved in buying and holding real estate with different holding periods are different. Taxes collected on the holding period of the real estate consist of value-added tax and individual income tax. For instance, housing with a holding period more than two years is exempt from the value-added tax (about 5.5% of the transaction price), and housing with a holding period more than five years is exempt from the individual income tax (20% of the difference between sale and purchase price or 1% of the transaction price). The samples in this paper are houses purchased and sold between 2017 and 2020. The maximum holding period is four years. In order to ensure the consistency and comparability of value-added tax, we divide the samples by whether the holding period is more than two years—i.e., into less than two years (0–2 years) and more than two years (2–4 years)—and use the two groups of samples to test the disposition effect, respectively. We expect that the short-term hold (0–2 years) has a more significant disposition effect. Considering the high value-added tax and surcharges payable on real estate transactions held for less than two years, sellers may take the 5.5% tax rate into account when estimating the unrealized returns. In the process of generating data, the return rate of less than 5.5% with a holding period of less than two years (8 quarters) is defined as a loss.

Columns (1)–(3) and (4)–(6) in Table 5 present the regression results of the samples with a holding period of 0–2 years and 2–4 years, respectively. All columns include control variables for house attributes, project characteristics, and seller information. Columns (1) and (4) control the time fixed effects at the calendar quarter level and the region fixed effects at the street level. Columns (2), (3), (5), and (6) are based on the OLS regression results, where columns (2) and (5) control the time fixed effects at the calendar quarter level and the region fixed effects at the street level, and the region fixed effects in columns (3) and (6) are controlled at the project level.

As stated in columns (1)–(3) in Table 5, the marginal effect in the probit model *Loss* in column (1) is −0.035, which is significant at the 1% level, suggesting that there is a negative correlation between *loss* and sell propensity, and that a *loss* property is 3.5% less likely to be sold than a *gain* property. Column (2) exhibits the OLS regression results, and the coefficient of *loss* is −0.035, which is also significant at the 1% level. Column (3) controls tighter region fixed effects at the project level, the absolute value of the *Loss* coefficient (−0.047) is greater than that in column (2), which is also significant at the 1% level. In the samples with less than two holding years, the *Loss* coefficients under different model settings are all significantly negative, indicating that investors have a lower propensity to sell houses that have lost value, which confirms the presence of the disposition effect in China’s real estate market. 

As the results in columns (4)–(6) in Table 5 show, under different model settings, the signs of the *Loss* coefficients are also significantly negative, but the absolute value of the *Loss* coefficient is smaller than that in columns (1)–(3). We believe that the possible reason for this result is that homeowners with longer holding periods are more likely to trade their homes for consumption purpose such as home replacement. For transactions motivated by consumption, investors are generally eager to realize assets to meet liquidity needs, and thus pay little attention to the gains or losses, which is in line with our expectations.

To sum up, we can draw two conclusions from Table 5: first, loss houses have a lower propensity to sell, which confirms the existence of the disposition effect in China’s real estate market; second, the disposition effect mainly appears in real estate transactions with shorter holding periods that are more likely for investment purposes. 

### 3.2. Robustness Analysis

To explain the reliability of the benchmark results in this paper, we next conduct a series of robustness analyses that may affect the estimated results. 

#### 3.2.1. Limiting the Scope of Purchase Dates

Our benchmark analysis mainly examines whether samples with a holding period of less than two years are listed for sale in 2019. This includes samples bought before 2019, such as a house purchased in 2018Q1 and listed for sale in 2019Q2. As we cannot observe samples bought in 2018Q1 but sold before 2019, to avoid the potential bias, we re-estimate the results by limiting the scope of purchase dates. To be specific, we remove samples purchased before 2019.

Table 6 shows the regression results after limiting the scope of purchase dates. After limiting the scope of purchase dates, the *Loss* coefficients of both the probit model and OLS model are still significantly negative at the 1% level, which is consistent with the benchmark results.

#### 3.2.2. Removing Samples of Unreliable and Extreme Returns

The definition of *Loss* (or *Return*), the core explanatory variable, relies on using the hedonic model (Equation (1)) to estimate the initial purchase price and the historical potential prices. The reliability of the hedonic model determines the reliability of the estimation results. Hence, we remove the neighborhood samples with *R*^2^ smaller than 0.8 when estimating the historical potential price of the house and test the disposition effect in the samples with a greater fit for the hedonic model. The results are reported in columns (1)–(3) of Table 7. It can be seen that, under different model settings, the *Loss* coefficients are all significant at the 1% level, which is consistent with the benchmark analysis results.

We also perform a robustness analysis by removing samples of extreme returns, i.e., those of returns outside the [−0.20, 0.20] interval. The results are reported in columns (4)–(6) of Table 7. The *Loss* coefficients are still significantly negative, and the absolute value of the *Loss* coefficient is even greater than that of the benchmark analysis. A possible reason is homeowners who do not sell their houses when returns surge are more likely to be consumers rather than investors. After removing extreme returns, the samples are mostly based on investment behavior, which will result in a more significant disposition effect.

#### 3.2.3. Adjusting Expected Returns

According to the definition of the sell propensity $Sell_{i,t}$, sellers decide whether to sell the house in period *t* + 1 by observing the house price in period *t*, which is consistent with the definition by Hong et al. [7]. Nevertheless, some scholars define the propensity to sell according to the current selling decision, i.e., sellers decide whether to sell assets in the *t* period by observing the return of the *t* period, such as Odean’s [2] “computing the proportion of gains realized” and a series of papers that apply Odean’s [2] method (Ben-David and Hirshleifer [11]). The definition of the sell propensity $Sell_{i,t}$ is shown in Equation (8), which is equivalent to considering price expectations on the basis of the definition in Equation (4). Sellers make a decision to sell in period *t* + 1 according to Equation (4) in period *t*, but sellers use the expected price in period *t* + 1 when evaluating returns. Now, we redefine the sell propensity according to Equation (8) and examine the testing results of the disposition effect when sellers make trading decisions based on return expectations. The regression results are shown in Table 8. Adjusting the return expectations does not affect the conclusion about the disposition effect.
(8)$$Sell_{i,t}=\{\begin{array}{cc} 1, & \begin{array}{cccc} Listed & in & period & t \end{array}\\ 0, & Otherwise \end{array}$$

#### 3.2.4. Different Testing Methods

To test the disposition effect, this paper uses the probit or OLS model for regression on returns and sell propensity. There are three other traditional methods to test the disposition effect: (1) comparing the proportion of gains realized to the proportion of losses realized; (2) comparing holding periods; (3) survival analysis.

Comparing the proportion of gains realized to the proportion of losses realized was first put forward by Odean [2]. Odean [2] divided the stocks in the investor’s account into four types on each selling day, according to whether they are sold and the gains and losses: realized gains, realized losses, paper gains, and paper losses, and then calculated and compared the proportion of gains realized (PGR) to the proportion of losses realized (PLR). We apply this method and find that the values of PLR–PGR in all periods are smaller than 0, which is consistent with the benchmark analysis results in this paper.

The basic idea of comparing the holding periods is that if investors are more inclined to hold loss assets, then under the same conditions, houses with paper losses should have a longer holding period than houses with paper gains (Shapria and Venezia [14]; Feng and Seasholes [15]). This method is not widely used due to sample selection problems. Therefore, when comparing the holding periods, we use the Heckman two-step selection method to reduce the impact of sample selection. In the first stage, we estimate the probit model shown in Equation (9) and compute the inverse Mill’s ratio (*IMR*). In the second stage we estimate Equations (10) and (11) including *IMR*, where the explained variable holding period (*Hold*) in Equation (10), and the logarithm of the holding period in Equation (11). Columns (1) and (2) of Table 9 give the results of Equations (10) and (11), respectively. It can be seen that, in Column (1), the coefficient of *Loss* is 0.356, which is significant at the 5% level; the coefficient of *Loss* in Column (2) is 0.058, which is significant at the 1% level, indicating that the holding period of loss houses is 0.356 months or 5.8% longer than that of gain houses, which is in line with disposition effect expectations.
(9)$$\mathrm{Pr}(Loss_{i,h,t}=1)=\Phi (\alpha_{0}+\alpha_{1}X_{i}+\alpha_{2}Z_{i,h,t}+\eta_{h}+\varphi_{t}+\varepsilon_{i,h,t})$$
(10)$$Hold_{i,h,t}=\alpha_{0}+\alpha_{1}Loss_{i,h,t}+\alpha_{2}X_{i}+\alpha_{3}Z_{i,h,t}+\alpha_{4}IMR+\eta_{h}+\varphi_{t}+\varepsilon_{i,h,t}$$
(11)$$\ln(Hold_{i,h,t})=\alpha_{0}+\alpha_{1}Loss_{i,h,t}+\alpha_{2}X_{i}+\alpha_{3}Z_{i,h,t}+\alpha_{4}IMR+\eta_{h}+\varphi_{t}+\varepsilon_{i,h,t}$$

In addition, Feng and Seacholes [15] applied a survival analysis to study the disposition effect, and the model is shown in Equation (12):(12)$$h_{i}(t|x_{i}(t))=h_{0}(t)\exp(\alpha_{0}+\alpha_{1}Loss_{i,h,t}+\alpha_{2}X_{i}+\alpha_{3}Z_{i,h,t}+\eta_{h}+\varphi_{t}+\varepsilon_{i,h,t})$$
where $h_{i}(t|x_{i}(t))$ is the risk ratio, which is the ratio of the probability that investors will not sell an asset until time *t*, and $h_{0}(t)$ is the benchmark risk ratio, which refers to the risk ratio when all variables are equal to 0. If $\alpha_{1}$ is significantly less than 0, we find the disposition effect. In this paper, when conducting a survival analysis to test the disposition effect and the exponential distribution, the Weibull distribution and the Cox model are adopted, respectively. The results are shown in Columns (3), (4), and (5) of Table 9. It can be observed that the coefficients of *Loss* are all significantly negative at the 1% level, which again verifies the existence of the disposition effect.

### 3.3. Heterogeneity Analysis 

Studies have shown that the disposition effect varies vastly across different groups of investors, such as those delineated by education level (Vaarmets et al. [16]). The dataset in this paper provides limit information about the investors. We measure the financial constraints faced by the investors according to mortgage information. Generally speaking, investors who use mortgages tend to face more financial constraints. For example, Genesove and Mayer [17] found that homeowners with a high loan-to-value ratio set a higher listing price, which is in line with the theory of financial constraints. Columns (1) and (2) of Table 10 give the probit regression results of the housing samples with and without a mortgage, respectively. It can be seen that the absolute value of the *Loss* coefficient (−0.045) of the housing samples with a mortgage is larger than that of the samples without a mortgage (−0.033), indicating that the disposition effect of investors with large financial constraints is more obvious. This finding agrees with the financial constraint theory proposed by Stein [18] that investors who are subject to tighter financial constraints are less able to improve housing or reinvest, and thus exhibit more pronounced disposition effects.

Some studies have also found that the investment environment will affect the level of the disposition effect. This paper marks the popularity of a project according to the average transaction cycle (TOM) of the samples from 2017 to 2020. The shorter the TOM, the higher the popularity of the project. We divide the projects into popular and unpopular ones according to the length of the average TOM to investigate the impact of project popularity on the disposition effect. From the probit regression results of the popular and unpopular project samples in columns (3) and (4) of Table 10, the results show that the disposition effect is more significant in popular neighborhoods, while unpopular projects do not show a significant disposition effect. The probable reason is that houses in popular projects have the benefit of good liquidity and loss sellers are in no hurry to sell, resulting in a greater disposition effect.

## 4. Further Analysis: Loss Aversion

The above section examines the disposition effect from the perspective of the influence of holding period returns (gains/losses) on sell propensity. Next, we refer to Genesove and Mayer [4] to study the loss aversion of real estate investors to further confirm the existence of the disposition effect. Loss aversion is an important concept to explain the disposition effect. The prospect theory coined by Kahneman and Tversky [19] believes that investors have an S-shaped utility function, and that investors are more sensitive to losses and will show the characteristics of loss aversion. The realization utility theory presented by Barberis and Xiong [20] and Ingersoll and Jin [21] also assumes that investors are loss averse, so loss aversion is often regarded as indirect evidence of the disposition effect. Loss aversion is mainly reflected in investors’ price expectations. Specifically, loss aversion makes investors hold high selling price expectations because they are unwilling to accept a loss. 

First, we follow the strategy of Genesove and Mayer [4] and use the listing price premium, which is the difference between the house listing price and the estimated price in the same period estimated by Equation (2), namely, $listpremium_{i,t}=\log listprice_{i,t}-\log\hat{p}rice_{i,t}$, to measure investors’ price expectations. Generally speaking, the larger the listing price premium, the higher the investor’s price expectations. The benchmark econometric regression model is shown in Equation (13), and the results are shown in column (1) of Table 11. The *Loss* coefficient is 0.036, which is significant at the 1% level, indicating that investors with loss houses tend to set a higher listing price.

We also consider the possible sample selection problem. First, we compute the Inverse Mill’s Ratio (IMR) based on the probit model shown in Equation (9), and then include the IMR into Equation (13). The results are shown in column (2) of Table 11. After adding the IMR, the coefficient of Loss does not change much and is still significant at the 1% level.

We further replace Loss with the return *R* computed by Equation (3) for regression. The econometric regression model is shown in Equation (14). The results are shown in column (3) of Table 11. The coefficient of *R* is significantly negative at the 1% level, suggesting that the greater the return, the lower the listing price. The results of Table 11 have verified the existence of loss aversion.
(13)$$listpremium_{i,h,t}=\alpha_{0}+\alpha_{1}Loss_{i,h,t}+\alpha_{2}X_{i}+\alpha_{3}Z_{i,h,t}+\eta_{h}+\varphi_{t}+\varepsilon_{i,h,t}$$
(14)$$listpremium_{i,h,t}=\alpha_{0}+\alpha_{1}R_{i,h,t}+\alpha_{2}X_{i}+\alpha_{3}Z_{i,h,t}+\eta_{h}+\varphi_{t}+\varepsilon_{i,h,t}$$

Since sellers may set listing prices for certain pricing strategies, such as high price and large discounts, a simple listing price premium does not necessarily reflect sellers’ price expectations. Therefore, we match the historical transaction data with the listing price data according to the house identification codes, obtain the final 7617 listing records, and use the difference between the transaction price and the estimated price $\log price_{i,t}-\log\hat{p}rice_{i,t}$, i.e., the transaction premium, to measure the price expectations of investors. We replace the transaction premium with the listing price premium in Equations (13) and (14) to examine the effect of holding period returns on the transaction premium. The results are shown in columns (4)–(6) of Table 11. The results of the transaction premium are consistent with the listing price premium. In summary, we have verified the existence of loss aversion among real estate investors, further providing empirical evidence for the disposition effect.

## 5. Discussion

This paper examines the relationship between property return and seller behavior. By matching more than 820,000 pieces of historical transaction data in the second-hand housing market of Beijing from 2012 to 2020 with more than 30,000 pieces of listing price data from January 2019 to December 2020, this paper estimates the purchase price and historical potential price of listed houses to compute the holding period return. We then estimate the relationship between holding period return and sell propensity. Consistent with evidence from developed countries, such as the United States (Genesove and Mayer [4]; Engelhardt [12]) and Singapore (Hong et al. [7]), we find that sellers in China’s real estate market exhibit the disposition effect.

This paper contributes to the literature on disposition effects in the real estate market. Case and Shiller [22] first found evidence of the disposition effect from interviews with homeowners in boom and post-boom real estate markets. Genesove and Mayer [4] used market data to examine the loss aversion of property sellers and found that loss aversion determined seller behavior in the housing market in Boston and that houseowners subject to expected losses set higher listing prices. However, their findings do not provide direct evidence of the disposition effect by examining unrealized return’s effect on propensity to sell. Hong et al. [7], using Singapore’s housing market data, were the first to test unrealized return’s effect on propensity to sell and provide direct evidence for the disposition effect. The research design of this paper draws on their paper; we find that sellers exhibit the disposition effect in China’s real estate market, which is consistent with evidence from developed countries, such as the United States (Genesove and Mayer [4]; Engelhardt [12]) and Singapore (Hong et al. [7]).

The findings of this paper also have important policy implications. On the one hand, the findings of the disposition effect and loss aversion can help policy-makers understand the behavior and psychology of property investors and provide directions for the formulation of policies related to limiting property speculation and stabilizing the housing market. On the other hand, traders with shorter holding periods are more likely to be motivated by investment, or more likely to be speculators, and the findings of this paper support a policy of linking taxes related to second home transactions to holding periods.

## 6. Conclusions

Our results show that loss properties have a lower sell propensity relative to gain properties, confirming the existence of the disposition effect in China’s real estate market. We find that the disposition effect is more pronounced in samples with holding periods shorter than two years. After removing the samples purchased before 2019, the unreliable community samples with R^2^ less than 0.8 in the hedonic model, and the extreme returns samples with holding period returns outside [−0.2, 0.2], we have found that the disposition effect remains significant. Using three other testing methods, i.e., comparing the proportion of gains realized to the proportion of losses realized, comparing holding periods, and making a survival analysis, we found that the disposition effect still exists.

Additionally, our heterogeneity analysis finds that the disposition effect is affected by the investor’s financial constraints and the popularity of the project. Specifically, we find that investors with financial constraints are more likely to show the disposition effect and the disposition effect is more significant in popular projects. Furthermore, we find that a property’s listing prices are inversely proportional to its holding period return, and that loss properties come along with higher listing transaction prices, confirming the existence of loss aversion and providing indirect evidence for the disposition effect.

## Figures and Tables

**Figure 1 behavsci-13-00055-f001:**
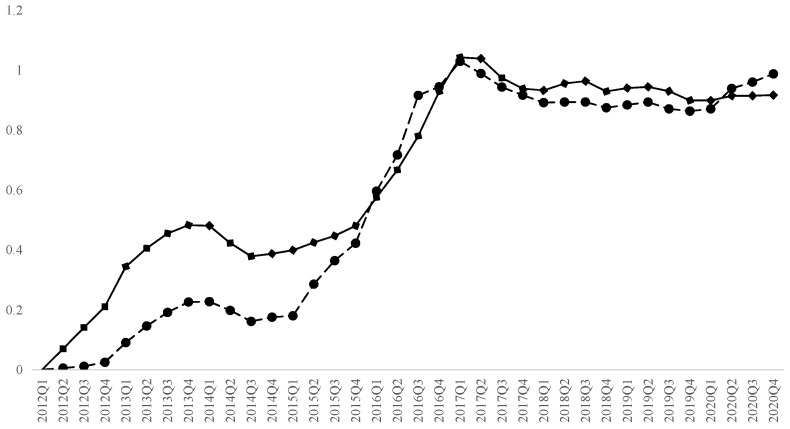
Historical potential prices and official housing prices. Note: The solid line in this figure shows the estimated historical potential price of the samples, and the dotted line shows second-hand housing price from the Beijing Bureau of Statistics. The base period for both curves is the first quarter of 2012.

**Table 1 behavsci-13-00055-t001:** Transaction data and descriptive statistics.

Variable	Description	Mean	SD	Max	Min
*List price total*	Total listing price	411.680	280.571	8800	10
*List price*	Listing price, unit: yuan/m^2^	49,780.204	24,229.086	199,901.039	10,000
*Price total*	Total transaction price	400.954	269.719	8000	12
*Price*	Transaction price, unit: yuan/m^2^	48,530.386	23,486.200	199,924.242	10,000
*Size*	House size, unit: m^2^	84.660	37.654	728	6
*Bedroom*	Number of bedrooms	2.034	0.779	9	1
*Living room*	Number of living rooms	1.163	0.504	6	0
*Bathroom*	Number of bathrooms	1.205	0.454	8	0
*Level1*	Level of the house. Two dummy variables to signify middle and high floors. *Level1* represents middle floors, and *Level2* represents higher floors.	0.379	0.485	1	0
*Level2*		0.329	0.470	1	0
*Floor*	Total floors of the building	13.511	7.880	42	1
*South*	Whether the direction of the house is south, 1 if yes, otherwise 0.	0.773	0.419	1	0
*TOM*	Time-on-the-market, the interval between transaction date and listing date.	76.813	189.386	3509	1

**Table 2 behavsci-13-00055-t002:** Listing data and descriptive statistics.

Variable	Description	Mean	SD	Max	Min
*Log. of list price*	Take the logarithm of the list price, unit: yuan/m^2^	10.988	0.410	12.398	9.302
*Size*	House size, unit: m^2^	82.883	40.538	744.95	9.64
*Bedroom*	Number of bedrooms	2.076	0.812	9	1
*Living room*	Number of living rooms	1.095	0.437	5	0
*Bathroom*	Number of bathrooms	1.200	0.480	6	0
*Level1*	Level of the house. Two dummy variables to signify middle and high floors. *Level1* represents middle floors, and *Level2* represents higher floors.	0.359	0.480	1	0
*Level2*		0.314	0.464	1	0
*Floor*	Total floors of the building	13.437	7.947	42	1
*South*	Whether the direction of the house is south, 1 if yes, otherwise 0.	0.791	0.407	1	0
*Mortgage*	Denote 1 if the listed house is mortgaged, and 0 if not.	0.299	0.458	1	0
*Distance*	Distance between its residential project and Tiananmen Square, unit: kilometer	14.735	10.595	111.297	0.418

**Table 3 behavsci-13-00055-t003:** Control variables and *R*^2^ results.

A: Results of control variables							
*Size*	*Level1*	*Level2*	*Floor*	*Bedroom*	*Livingroom*	*Bathroom*	*South*
−0.003	0.010	−0.007	0.001	0.030	0.012	0.015	0.048
[−47.05]	[13.81]	[−7.22]	[0.63]	[21.29]	[12.15]	[6.41]	[25.58]
B: *R*^2^ distribution							
Mean	P1	P10	P25	P50	P75	P90	P99
0.907	0.481	0.792	0.892	0.943	0.967	0.979	0.992

Note: Panel A of Table 3 shows the mean values and *t*-test results of the control variables of 3623 neighborhoods estimated by Equation (1), and the t-statistics are shown in []; Panel B displays the distribution of *R*^2^, where P1, P10, P25, P50, P75, P90, and P99 represent the 1%, 10%, 25%, 50%, 75%, 90%, and 99% quantile, respectively.

**Table 4 behavsci-13-00055-t004:** Data derived process.

Variable	Period *t*_0_	Period *t*_0_ + 1	...	Period *t* − 1	Period *t*
Sell propensity	0	0	...	0	1
Unrealized return	$R_{0|t_{0}}$	$R_{1|t_{0}}$	...	$R_{t-1|t_{0}}$	$R_{t|t_{0}}$
Holding period	0	1	...	*t*−1	*t*
Housing attributes	*X_i_*	*X_i_*	...	*X_i_*	*X_i_*

**Table 5 behavsci-13-00055-t005:** Testing results of the disposition effect.

	0~2 Holding Years			2~4 Holding Years		
	(1)	(2)	(3)	(4)	(5)	(6)
	Probit	OLS	OLS	Probit	OLS	OLS
*Loss*	−0.035 ***	−0.035 ***	−0.047 ***	−0.025 ***	−0.017 ***	−0.020 ***
	(0.007)	(0.008)	(0.009)	(0.004)	(0.004)	(0.005)
*Purchase Price*	0.017	0.018	0.058 **	0.021 ***	0.019 ***	0.031 **
	(0.012)	(0.012)	(0.023)	(0.007)	(0.006)	(0.012)
*LnTOM*	−0.008 **	−0.007 **	−0.007 *	0.002	0.002	0.001
	(0.003)	(0.003)	(0.004)	(0.002)	(0.002)	(0.002)
*Distance*	−0.002	−0.002		−0.001	−0.001	
	(0.0062)	(0.002)		(0.001)	(0.001)	
*Hold Period*	0.066 ***	0.069 ***	0.076 ***	0.034 ***	0.028 ***	0.031 ***
	(0.001)	(0.001)	(0.001)	(0.001)	(0.001)	(0.001)
*Mortgage*	−0.073 ***	−0.069 ***	−0.068 ***	−0.033 ***	−0.030 ***	−0.037 ***
	(0.006)	(0.005)	(0.007)	(0.003)	(0.003)	(0.003)
Other Control Variables	Yes	Yes	Yes	Yes	Yes	Yes
Quarterly Fixed Effects	Yes	Yes	Yes	Yes	Yes	Yes
Street Fixed Effects	Yes	Yes	No	Yes	Yes	No
Project Fixed Effects	No	No	Yes	No	No	Yes
Sample Size	35,389	36,724	36,549	59,202	61,829	61,778
R^2^	0.167	0.255	0.269	0.215	0.343	0.349

Note: probit models show the marginal effect of variables; standard errors are given in ( ), and all standard errors are clustered at the project level; *, **, and *** represent the significance levels of 10%, 5%, and 1%, respectively; control variables involve house attribute variables such as types of houses (number of bedrooms, living rooms, and bathrooms), house size, house orientation (south-facing or not), floor (two dummy variables of middle floors and higher floors), and total floors; R^2^ refers to pseudo R^2^ in probit models and adjusted R^2^ in OLS models.

**Table 6 behavsci-13-00055-t006:** Robustness analysis: limit the scope of purchase dates.

	(1)	(2)	(3)
	Probit	OLS	OLS
*Loss*	−0.040 ***	−0.037 ***	−0.045 ***
	(0.011)	(0.011)	(0.013)
Control Variables	Yes	Yes	Yes
Quarterly Fixed Effects	Yes	Yes	Yes
Street Fixed Effects	Yes	Yes	No
Project Fixed Effects	No	No	Yes
Sample Size	14,604	16,812	16,695
R^2^	0.152	0.322	0.341

Note: probit models show the marginal effect of variables; standard errors are given in ( ), and all standard errors are clustered at the project level; *, ** and, *** represent the significance levels of 10%, 5%, and 1%, respectively; control variables involve house attribute variables such as types of houses (number of bedrooms, living rooms, and bathrooms), house size, house orientation, floor (two dummy variables of middle floors and higher floors), and total floors, project characteristic variables such as distance to Tiananmen Square and project liquidity, as well as trader information variables such as initial purchase price, holding period, and mortgage; R^2^ refers to pseudo R^2^ in probit models and adjusted R^2^ in OLS models.

**Table 7 behavsci-13-00055-t007:** Robustness analysis: remove samples of unreliable and extreme returns.

	Remove *R*^2^ < 0.8 Neighborhood Samples			Samples of Returns Inside the [−0.20, 0.20] Interval		
	(1)	(2)	(3)	(4)	(5)	(6)
	Probit	OLS	OLS	Probit	OLS	OLS
*Loss*	−0.028 ***	−0.026 ***	−0.038 ***	−0.036 ***	−0.036 ***	−0.048 ***
	(0.008)	(0.008)	(0.009)	(0.007)	(0.008)	(0.009)
Control Variables	Yes	Yes	Yes	Yes	Yes	Yes
Quarterly Fixed Effects	Yes	Yes	Yes	Yes	Yes	Yes
Street Fixed Effects	Yes	Yes	No	Yes	Yes	No
Project Fixed Effects	No	No	Yes	No	No	Yes
Sample Size	29,045	30,152	29,992	33,675	34,973	34,771
R^2^	0.168	0.256	0.269	0.168	0.257	0.269

Note: probit models show the marginal effect of variables; standard errors are given in ( ), and all standard errors are clustered at the project level; *, **, and *** represent the significance levels of 10%, 5%, and 1%, respectively; control variables involve house attribute variables such as types of houses (number of bedrooms, living rooms, and bathrooms), house size, house orientation, floor (two dummy variables of middle floors and higher floors), and total floors, project characteristic variables such as distance to Tiananmen Square and project liquidity, as well as trader information variables such as initial purchase price, holding period, and mortgage; R^2^ refers to pseudo R^2^ in probit models and adjusted R^2^ in OLS models.

**Table 8 behavsci-13-00055-t008:** Robustness analysis: adjust expected returns.

	(1)	(2)	(3)
	Probit	OLS	OLS
*Loss*	−0.024 ***	−0.023 ***	−0.035 ***
	(0.007)	(0.007)	(0.008)
Control Variables	Yes	Yes	Yes
Quarterly Fixed Effects	Yes	Yes	Yes
Street Fixed Effects	Yes	Yes	No
Project Fixed Effects	No	No	Yes
Sample Size	43,529	44,852	44,720
R^2^	0.153	0.233	0.242

Note: probit models show the marginal effect of variables; standard errors are given in ( ), and all standard errors are clustered at the project level; *, **, and *** represent the significance levels of 10%, 5%, and 1%, respectively; control variables involve house attribute variables such as types of houses (number of bedrooms, living rooms, and bathrooms), house size, house orientation, floor (two dummy variables of middle floors and higher floors), and total floors, project characteristic variables such as distance to Tiananmen Square and project liquidity, as well as trader information variables such as initial purchase price, holding period, and mortgage; R^2^ refers to pseudo R^2^ in probit models and adjusted R^2^ in OLS models.

**Table 9 behavsci-13-00055-t009:** Robustness analysis: different testing methods.

	(1)	(2)	(3)	(4)	(5)
	OLS	OLS	Exponential	Weibull	Cox
*Loss*	0.356 **	0.058 ***	−0.070 ***	−0.095 ***	−0.118 ***
	(0.148)	(0.021)	(0.012)	(0.016)	(0.018)
Control Variables	Yes	Yes	Yes	Yes	Yes
Quarterly Fixed Effects	Yes	Yes	Yes	Yes	Yes
Street Fixed Effects	No	No	Yes	Yes	Yes
Project Fixed Effects	Yes	Yes	No	No	No
Sample Size	17,487	17,487	15,059	15,059	15,059
Pseudo R^2^	0.336	0.267			0.004

Note: probit models show the marginal effect of variables; standard errors are given in ( ), and all standard errors are clustered at the project level; *, **, and *** represent the significance levels of 10%, 5%, and 1%, respectively; control variables involve house attribute variables such as types of houses (number of bedrooms, living rooms, and bathrooms), house size, house orientation, floor (two dummy variables of middle floors and higher floors), and total floors, project characteristic variables such as distance to Tiananmen Square and project liquidity, as well as trader information variables such as initial purchase price, holding period, and mortgage; columns (1) and (2) also include the inverse Mill’s ratio (IMR) calculated according to Equation (9).

**Table 10 behavsci-13-00055-t010:** Heterogeneity analysis.

	Mortgage		Project Popularity	
	Without a Mortgage	With a Mortgage	Unpopular	Popular
	(1)	(2)	(3)	(4)
*Loss*	−0.033 ***	−0.045 ***	−0.015	−0.045 ***
	(0.008)	(0.013)	(0.012)	(0.009)
Control Variables	Yes	Yes	Yes	Yes
Quarterly Fixed Effects	Yes	Yes	Yes	Yes
Street Fixed Effects	Yes	Yes	Yes	Yes
Sample Size	26,055	9315	12,097	23,272
Pseudo R^2^	0.154	0.236	0.187	0.168

Note: probit models show the marginal effect of variables; standard errors are given in ( ), and all standard errors are clustered at the project level; *, **, and *** represent the significance levels of 10%, 5%, and 1%, respectively; control variables involve house attribute variables such as types of houses (number of bedrooms, living rooms, and bathrooms), house size, house orientation, floor (two dummy variables of middle floors and higher floors), and total floors, project characteristic variables such as distance to Tiananmen Square and project liquidity, as well as trader information variables such as initial purchase price, holding period, and mortgage.

**Table 11 behavsci-13-00055-t011:** Loss aversion.

	Listing Price Premium			Transaction Premium		
	(1)	(2)	(3)	(4)	(5)	(6)
*Loss*	0.036 ***	0.037 ***		0.031 ***	0.032 ***	
	(0.003)	(0.003)		(0.004)	(0.004)	
*R*			−0.526 ***			−0.446 ***
			(0.034)			(0.045)
IMR		0.021			0.035	
		(0.020)			(0.024)	
Control Variables	Yes	Yes	Yes	Yes	Yes	Yes
Quarterly Fixed Effects	Yes	Yes	Yes	Yes	Yes	Yes
Street Fixed Effects	Yes	Yes	Yes	Yes	Yes	Yes
Sample Size	17,505	16,975	17,505	7617	7484	7617
R^2^	0.146	0.147	0.206	0.096	0.096	0.148

Note: standard errors are given in ( ), and all standard errors are clustered at the neighborhood level; *, **, and *** represent the significance levels of 10%, 5%, and 1%, respectively; control variables involve house attribute variables such as types of houses (number of bedrooms, living rooms, and bathrooms), house size, house orientation (south-facing or not), floor (two dummy variables of middle floors and higher floors), and total floors, neighborhood characteristic variables such as mortgage, distance to Tiananmen Square, total purchase price, and neighborhood liquidity, as well as trader information variables such as initial purchase price, holding period, and mortgage.

## Data Availability

Not applicable.

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
