# Peer review of "Does Property Return Affect Seller Behavior? An Empirical Study of China’s Real Estate Market"

_behavsci, 2023, doi:10.3390/bs13010055_

Round 1

Reviewer 1 Report

The paper is very well written and the topic is relevant. My two  suggestions are below.

1. The paper examines property return and seller behavior - which would be valuable to inform a seller for setting pricing for different housing typologies. Yet, the paper is not written with the general audience in mind, and fails to provide real-life application of these findings in the discussion section. I would suggest explaining how these findings can support on sellers decision-making. 

2. The introduction is succinct but could be expanded on to discuss more on the gaps in not only literature but in practice and be tied into the real-life need for the work (similar to my first suggestion). 

Author Response

  1. The paper examines property return and seller behavior - which would be valuable to inform a seller for setting pricing for different housing typologies. Yet, the paper is not written with the general audience in mind, and fails to provide real-life application of these findings in the discussion section. I would suggest explaining how these findings can support on sellers decision-making.

We appreciate your comments, and we realize that the introduction and discussion sections of this paper are missing much of the practical and policy implications of the paper, so we have added "Contributions of this paper" to the introduction as follows.

“This paper has three main contributions: Firstly, previous empirical studies on the disposition effect have mainly focused on the stock market, and there is less research on the disposition effect in the real estate market, this paper provides evidence of the disposition effect in the Chinese real estate market. Secondly, this paper has provided an new empirical method to test the effect of holding period on the propensity to sell, which can provide new ideas for research related to the propensity to sell of houses. Thirdly, the findings of this paper help to understand the behavior and psychology of property investors and provide direction for policy makers to limit property speculation and stabilize the housing market.”

  1. The introduction is succinct but could be expanded on to discuss more on the gaps in not only literature but in practice and be tied into the real-life need for the work (similar to my first suggestion).

In the Discussion section, we have added the relevance and policy implications of the conclusions of this paper as follows.

“The findings of this paper also have important policy implications. On the one hand, the findings of the disposition effect and loss aversion can help understand the behavior and psychology of property investors and can provide directions for the formulation of policies related to limiting property speculation and stabilizing the housing market. On the other hand, traders with shorter holding periods are more likely to be motivated by investment, or more likely to be speculators, and the findings of this paper support a policy of linking taxes related to second home transactions to holding periods.”

Reviewer 2 Report

This manuscript is a clever and well-structured piece of real estate research. It is a proof of existence of Chinese RE market and a proof of research talent using Big Data based market analysis. Authors invested tremendous work to obtain their result which is meaningful and robust enough.

Based on their huge and unique dataset other investigations and tests could be made; Authors, as auxiliary conclusions have envisioned some of them.

Only recommendation is that Authors should proof the adequacy of used house characteristic variables, whether the set of variables is a full representation of the market value. The used holding period is relative short, especially in the case of investment properties. Authors may state that new, western-type market is very young and no longer investment period can be observed.

Author Response

  • Authors should proof the adequacy of used house characteristic variables, whether the set of variables is a full representation of the market value.

This is a good question, and the reliability of the estimate of housing market value is the basis of this paper. Indeed, we can proof the adequacy of used house characteristic variables.

In order to estimate the historical potential price of the house. We does not use the historical average price of the project as the historical potential price, but to estimate the historical potential prices project by project.

As shown in Table 3, the mean value of R2 of 3623 regression equations has hit 0.907, and the 10% quantile has reached 0.792, which shows that the hedonic price model has good explanatory power, and the set of variables is a full representation of the market value. In addition other unobserved variables have a limited impact.

  • The used holding period is relative short, especially in the case of investment properties. Authors may state that new, western-type market is very young and no longer investment period can be observed.

Thank you for your question about the holding period. The samples in this paper are houses purchase and sold between 2017 and 2020. The maximum holding period is 4 years. Admittedly, we could not obtain a longer sample of holding times due to data limitations, which limits our ability to further investigate seller behavior based on consumption motives. However, since the main purpose of this paper is to examine the disposition effect of sellers in the real estate market, which is mainly motivated by investment rather than consumption. A series of studies suggest that shorter holding periods are more likely to be out of investment rather than consumption motives (Ben-David and Hirsheifer, 2012; Hong et al., 2022). So we argue the relative short holding period is is sufficient for the purpose of our study.
